# Verte-Box: A Novel Convolutional Neural Network for Fully Automatic Segmentation of Vertebrae in CT Image

**Bing Li** [1,2,*]**, Chuang Liu** [1]**, Shaoyong Wu** [1] **and Guangqing Li** [1]

[1] School of Automation, Harbin University of Science and Technology, Harbin 150080, China; 1920500002@stu.hrbust.edu.cn (C.L.); 2020500046@stu.hrbust.edu.cn (S.W.); 2020510117@stu.hrbust.edu.cn (G.L.)

[2] Heilongjiang Provincial Key Laboratory of Complex Intelligent System and Integration, School of Automation, Harbin University of Science and Technology, Harbin 150080, China

[*] Correspondence: libing@hrbust.edu.cn

**Abstract:** Due to the complex shape of the vertebrae and the background containing a lot of interference information, it is difficult to accurately segment the vertebrae from the computed tomography (CT) volume by manual segmentation. This paper proposes a convolutional neural network for vertebrae segmentation, named Verte-Box. Firstly, in order to enhance feature representation and suppress interference information, this paper places a robust attention mechanism on the central processing unit, including a channel attention module and a dual attention module. The channel attention module is used to explore and emphasize the interdependence between channel graphs of low-level features. The dual attention module is used to enhance features along the location and channel dimensions. Secondly, we design a multi-scale convolution block to the network, which can make full use of different combinations of receptive field sizes and significantly improve the network's perception of the shape and size of the vertebrae. In addition, we connect the rough segmentation prediction maps generated by each feature in the feature box to generate the final fine prediction result. Therefore, the deep supervision network can effectively capture vertebrae information. We evaluated our method on the publicly available dataset of the CSI 2014 Vertebral Segmentation Challenge and achieved a mean Dice similarity coefficient of 92.18 ± 0.45%, an intersection over union of 87.29 ± 0.58%, and a 95% Hausdorff distance of 7.7107 ± 0.5958, outperforming other algorithms.

**Keywords:** CT image; vertebrae segmentation; feature enhancement; multi-scale; convolutional neural network

## 1. Introduction

Automatic vertebrae segmentation from medical images is critical for spinal disease diagnosis and treatment, e.g., assessment of spinal deformities, surgical planning, and postoperative assessment; computed tomography (CT) is one of the most commonly used imaging methods in clinical practice [1], and the convolutional neural network has become the best choice for processing such images.

In practice, vertebrae segmentation from volumetric CT image suffers from the following challenges:

- Inter-class similarity: Shape and appearance similarities appear in the neighboring vertebrae from the sagittal view. It is difficult to distinguish the first lumbar vertebra (L1) and the second lumbar vertebra (L2) (as shown in Figure 1a);
- Unhealthy vertebrae, such as deformity or lesions (as shown in Figure 1b);
- Interference information: there are several soft tissues whose gray-scale is similar to the vertebrae area (as shown in Figure 1c,d).

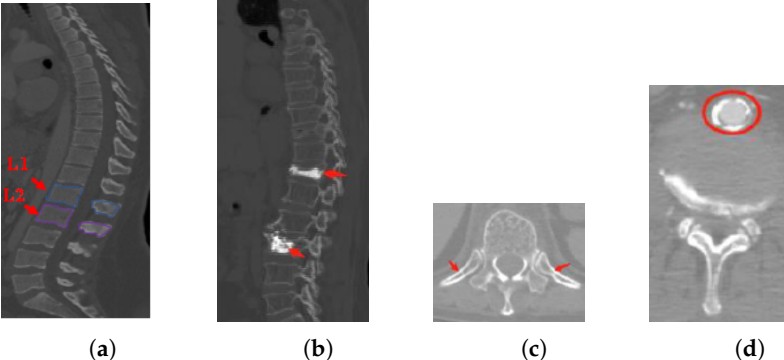

**Figure 1.** The challenges of vertebrae segmentation. (**a**) Inter-class similarity. (**b**) Unhealthy vertebrae. (**c**,**d**) Interference information.

Vertebrae segmentation has been approached predominantly as a model-fitting problem using statistical shape models (*SSM*) [2,3] or active contour methods [4,5] in the early stage. However, most traditional methods are only suitable for high-quality CT images in which the vertebrae are healthy, and they cannot perform well in the complex scenario shown in Figure 1. Subsequently, machine learning has been widely used. Chu et al. [6] detected the center of the vertebral bodies using random forest and Markov random fields and then used these centers to obtain fixed-size regions of interest (*ROI*), in which vertebrae were segmented using random forest voxel classification. Korez et al. [7] used a convolution neural network to generate a probability map of vertebrae, and these maps were used to guide the deformable model to segment each vertebra. Although machine learning methods outperform the traditional approaches in speed and efficiency, they have no obvious advantage in segmentation accuracy.

More recently, the deep learning model *U-Net* [8] has achieved great success in the task of medical image segmentation, which has become one of the most popular models in this field. Kim et al. [9] proposed a deep learning algorithm to detect ankle fractures from X-rays and CT scans and achieved quite good results.Their work proved the powerful ability of deep learning in AI-assisted diagnosis and treatment. Moreover, Holbrook et al. [10] and Yogananda et al. [11] proposed two improved U-Net networks for tumor segmentation respectively. In order to improve the segmentation accuracy, many models based on a cascade structure were proposed. Sekuboyina et al. [12] presented a two-stage lumbar vertebrae segmentation algorithm, which first extracted the ROI of lumbar vertebrae using the multi-layer perceptron (*MLP*) and then performed instance segmentation within these regions. Similarly, Janssens et al. [13] segmented the five lumbar vertebrae using two-cascade *3D U-Net*. However, these methods are not designed for spinal CT images that have a varying number of visible vertebrae and can only be used to segment the lumbar vertebrae. In order to solve the problem of a prior unknown number of vertebrae, Lessmann et al. [14] proposed an iterative vertebrae instance segmentation method, which slides a window over the images until a complete vertebra is visible, and then performed instance segmentation and saved it to the memory module. This process was repeated until all fully visible vertebrae were segmented. Unfortunately, there is a deficiency in this method, which requires a lot of memory during training.

In this study, we aim to propose a novel neural network applied to the vertebrae segmentation task. Firstly, the size and position of the vertebrae are not fixed in CT volumes. The $3 \times 3$ filter is too small to extract global information. Simply expanding the kernel size will quickly increase the number of parameters. We explore using a different combination of receptive field sizes to improve the network perception concerning the shape and size of vertebrae. Secondly, treating the representations on the different spatial positions and channels in the feature map equally will result in a large amount of computation cost and lower segmentation accuracy. At the same time, in order to enhance feature representation and suppress interference information, we introduce an attention mechanism to explore

and emphasize the interdependencies between channel maps. Finally, a coarse-to-fine strategy is employed in the prediction stage: every decoded feature will produce a coarse segmentation and then concatenate them to generate the final fine prediction.

## 2. Materials and Methods

### 2.1. Architecture

The architecture of Verte-Box is depicted in Figure 2. The network can be divided into three stages: feature extraction, middle-processing, and prediction. In the first stage, a semantic feature extractor is used to generate the semantic image representation by five consecutive max-pooling and multi-scale convolutions; every time, the number of channels is doubled, and the feature map size is reduced by half. In the middle-processing stage, the channel attention module and dual attention module contained in the central processing unit are responsible for enhancing feature representation and suppressing interference information. In the prediction stage, every feature in the feature box will produce a coarse segmentation and then concatenate them to generate the final fine prediction.

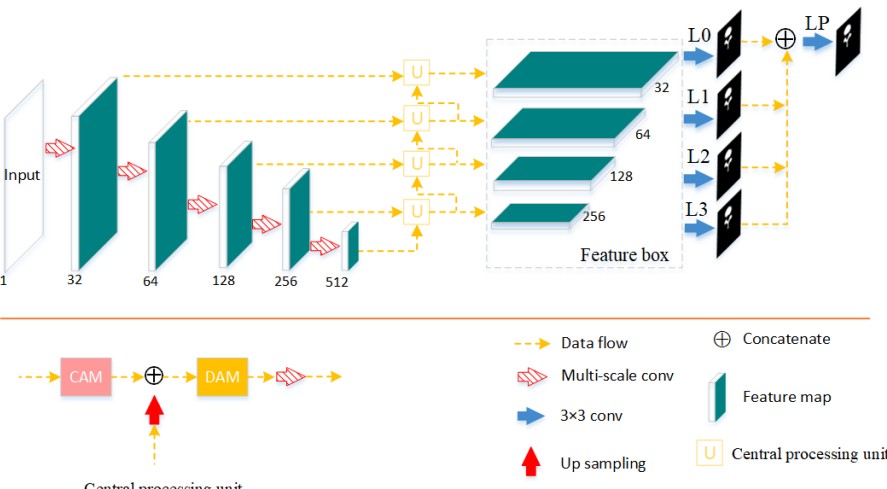

**Figure 2.** The top is a schematic drawing of our Verte-Box architecture. The bottom left is the central processing unit which a contains channel attention module (CAM) and dual attention module (DAM). L0–L3 are auxiliary segmentation loss terms, and Lp is a main loss. 'Verte' denotes vertebrae, and 'box' denotes feature box.

### 2.2. Attention Mechanism

In this work, we propose a new dual attention mechanism. As illustrated in Figure 3, this attention mechanism uses a parallel structure: the top is the position attention module (*PAM*), and the bottom is the channel attention module (*CAM*). In order to make the two attention modules complementary, we aggregate the features from these two attention modules. Concretely, we perform an element-wise sum to accomplish feature fusion. Given an input feature map $X \in R^{C \times H \times W}$, the overall attention mechanism can be summarized as

$$\widehat{X} = F_C(X) + F_P(X) \tag{1}$$

where $F_C$ denotes *CAM*, and $F_P$ denotes *PAM*.

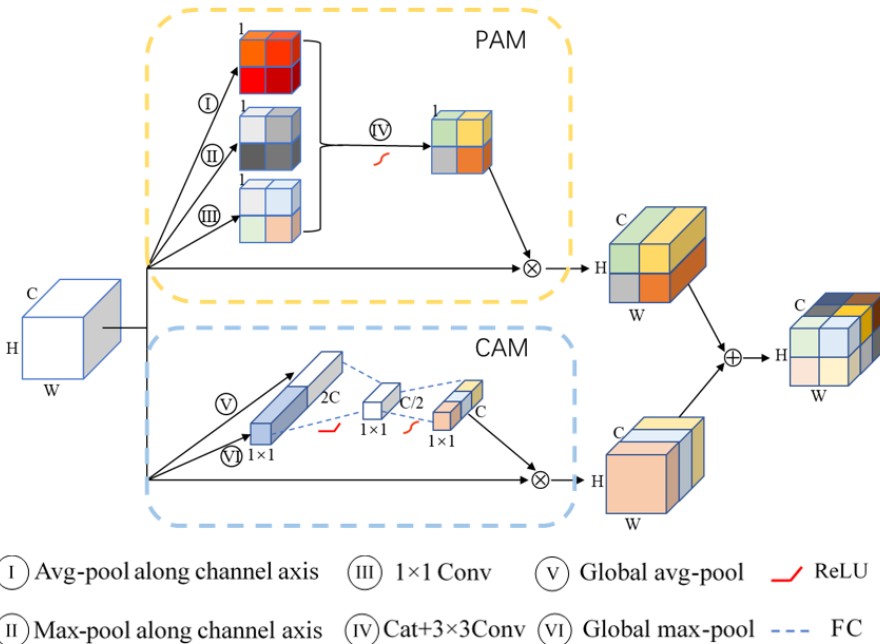

**Figure 3.** Illustrating the pipeline of dual attention module (DAM). (1) form the PAM in the yellow dashed box. (2) form the CAM in the blue dashed box. Weights are represented in different colors.

For channel attention module, given an input $X$, a convolutional operator [15] can be written as follows:

$$u_i = k_i * X = \sum_{s=1}^{C} k_i^s * x^s \tag{2}$$

where $*$ denotes convolution $X = \left[x^1, x^2, \cdots, x^C\right]$, $k_i = \left[k_i^1, k_i^2, \cdots, k_i^C\right]$, $k_i$ refers to the parameters of the $i$-th kernel, and $u_i \in R^{H \times W}$ denotes the i-th channel of output feature map. $k_i^s$ is a 2D spatial kernel representing a single channel $k_i$ that acts on the corresponding channel of $X$. It can be found that convolution is a summation through all channels; channel dependencies are implicitly embedded in every kernel $k_i$. Meanwhile, affected by the receptive field of each filter, the channel relationships modeled by convolution are local. To alleviate those issues, we need to do two things. First, we need to provide the network with access to global information of each channel. Concretely, a channel descriptor $z_C \in R^{2C}$ is generated by global average pooling and max pooling:

$$z_C = \begin{cases} \frac{1}{H \times W} \sum_{i=1}^{H} \sum_{j=1}^{W} x^s(i,j) & 1 \leq s \leq C \\ \max_{i,j} x^s(i,j) & C+1 \leq s \leq 2C \end{cases} \tag{3}$$

Second, in order to explicitly capture channel-wise dependencies, a simple gating mechanism (a bottleneck with two fully connected layers) with a sigmoid activation is applied:

$$S = \sigma(W_2 \delta(W_1 z_c)) \tag{4}$$

where $\delta$ refers to the *ReLU* function, $W_1 \in R^{(C/4) \times C}$, and $W_2 \in R^{C \times (C/4)}$.

Finally, the output $F_c$ is obtained by rescaling $X$ with the activations $S$:

$$F_C(X) = X \bigotimes S \tag{5}$$

where $\otimes$ refers to channel-wise multiplication.

Similarly, for position attention module, the network needs global information of different channels. Three position descriptors $z_p^i \in R^{H \times W}$, $i \in \{1, 2, 3\}$ are generated by average pooling, max pooling along the channel dimension, and a $1 \times 1$ convolution operation:

$$z_P^i = \begin{cases} \max\limits_{s=1,2,\cdots,C} x^s & i = 1 \\ \sum_{s=1}^{C} x^s & i = 2 \\ k^{1 \times 1} * X & i = 3 \end{cases} \tag{6}$$

Finally, the output $F_P$ is obtained by

$$F_P(X) = X \otimes \sigma(k^{3 \times 3} * z_P) \tag{7}$$

where $z_P = \left[ z_P^1, z_P^2, z_P^3 \right]$.

### 2.3. Multi-Scale Convolution

We design a new multi-scale convolution whose structure is shown in Figure 4a. The cascading of one $3 \times 3$ convolution and two *Res2net* modules can further expand the receptive field of the network. Meanwhile, the *Res2net* [16] module was used to extract multi-scale features to obtain the local and global information. Figure 4b shows the structure of the *Res2net* module. After a $1 \times 1$ convolution, the input feature map is split into $S$ feature map subsets, denoted by $X_i \in R^{(C/S) \times H \times W}$, $\{1 \leq i \leq S\}$. The feature subset $X_i$ is added with the output $Y_{i-1}$ and then fed into layer. In order to reduce parameters, $Y_1$ is obtained from $X_1$ by identity mapping. The output of the *Res2net* module, the $Y$, can be represented by Equation (5):

$$Y_i = \begin{cases} X_i & i = 1 \\ f^{3 \times 3}(x_i) & i = 2 \\ f^{3 \times 3}(x_i + Y_{i-1}) & 2 < i \leq S \end{cases} \tag{8}$$

$$Y = f^{1 \times 1}(\{Y_1, Y_2, \cdots, Y_C, \}) \tag{9}$$

Notice that $Y_i$ is obtained by $X_i$ after a $3 \times 3$ convolution, its receptive field is larger than $X_i$, and benefiting from the hierarchical residual-like structure, $Y_i$ could potentially receive feature information from all feature subsets $\{X_j, j \leq i\}$. Due to the combinatorial explosion effect, Y contains a different number and different combination of receptive field scales of all the subsets $\{X_j, 1 \leq j \leq S\}$. The hyperparameter of $S$ is related to the number of channels, and in this work, the first three layers of the downsampling and the last two layers of the up-sampling, $S$, are set to 4. With the increase of the number of channels, the last two layers of the down-sampling and the first two layers of the up-sampling, $S$, are set to 6.

### 2.4. Deep Supervision

In the training process, we use deep supervision; $\lambda_i$ and $\lambda_P$ are the weights of each loss term. For each auxiliary segmentation term $L_i$ and the final output $L_P$, we use the *Dice-loss*, which performs well in eliminating the impact of class imbalance. The loss is calculated by

$$L = \sum_{i=1}^{4} \lambda_i L_i + \lambda_P L_P \tag{10}$$

$$L(y, P) = 1 - \frac{1}{C} \sum_{j}^{C} \frac{2 \sum_i^N y_{ij} P_{ij}}{\sum_i^N y_{ij} + \sum_1^N p_{ij}} \tag{11}$$

where $C$ is the number of classes, and $y$ and $P$ denote the one-hot encoding of ground truth and the predicted result, respectively. $N$ is the number of pixels in each image. In the test process, we choose the final output $L_P$ as our prediction result.

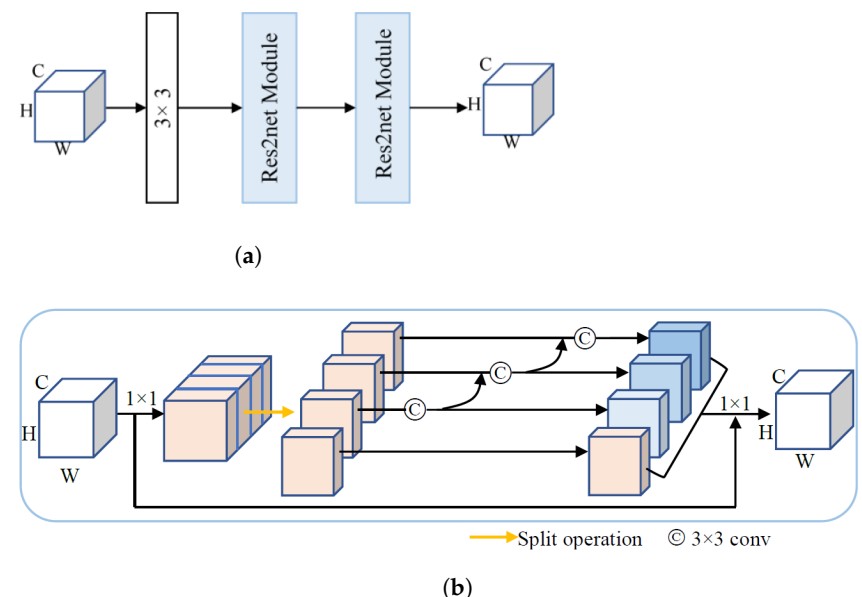

**(a)**

**(b)**

**Figure 4.** Multi-scale convolution. (**a**) Multi-scale convolution. (**b**) Res2net module structure, where circle C denotes 3 × 3 convolution.

## 3. Results

### 3.1. Dataset and Implementation Details

We evaluated our model on a public dataset provided by the *CSI 2014 Vertebral Segmentation Challenge* [17]. The dataset contains 20 CT scans. Those scans cover the entire thoracic and lumbar spine, the in-plane resolution is between 0.31 and 0.45 mm, and the slice thickness is 1 mm. In this paper, all scans are resampled to 1 mm × 1 mm × 1 mm, 14 scans are used as a training set, and the remaining 6 scans are used as a test set. We sliced the training set scans along the axial plane, and the image size was resized to 256 × 256. There are a total number of 5980 slice images which are divided into a training set and validation set by a 4:1 ratio. For data augmentation during training, the images were randomly rotated (±0.1 rad) and flipped; we also employed contrast adjustment and Gaussian noise. The loss weights $\lambda_i$ and $\lambda_p$ are all set to 1. We trained the network using the *AdamW* optimizer with a learning rate of 0.001 and a mini-batch of 16 on an Intel(R) Xeon(R) workstation with an NVIDIA RTX 3070 running Ubuntu Linux. During training, if the validation set Dice value does not rise for 10 consecutive epochs, the learning rate is halved. The maximum number of training epochs is 100. All implementations were done in *Pytorch 1.8.0*.

### 3.2. Metrics

In terms of the segmentation accuracy, the most commonly used evaluation metrics include the Dice similarity coefficient (*DSC*), intersection over union (*IOU*), and 95% Hausdorff Distance (95% HD).

Given the set *A* and another set *B*, their Dice coefficients can be defined as follows. The Dice coefficient measures the overlap between segmentation results and ground truth, which is not affected by the class-imbalance. Dice values range from 0 to 1, with values closer to 1 indicating better segmentation.

$$DSC(A, B) = \frac{2 \times |A \cap B|}{|A| + |B|} \tag{12}$$

Similar to *DSC*, *IOU* is also used to measure the overlap between two sets, as shown in Equation (13):

$$IOU(A, B) = \frac{|A \cap B|}{|A \cup B|} \tag{13}$$

The 95% HD: The maximum Hausdorff distance is the maximum distance of set A to the nearest point in the other set *B*, and it is a good reflection of the difference between the split result and the *GT* boundary, defined as

$$H(A, B) = \max(h(A, B), h(B, A)) \tag{14}$$

$$h(A, B) = \max_{a \in A}\left\{ \min_{b \in B}\{d(a, b)\} \right\} \tag{15}$$

$$h(B, A) = \max_{b \in B}\left\{ \min_{a \in A}\{d(b, a)\} \right\} \tag{16}$$

where *d* is the Euclidean distance between the pixels *a* and *b*, so the unit of 95% HD is the distance between two pixels. The 95% HD is similar to the maximum *HD*, and it is based on the calculation of the 95th percentile of the distance between boundary points in *A* and *B*.

### 3.3. Experimental Data Analysis

We compare our model with four related state-of-the-art methods, *SegNet* [18], *U-Net*, Attention U-Net (*AU-Net*) [19], and *U-Net++* [20]. The segmentation results visualization is shown in Figure 5, and the quantitative segmentation results are shown in Table 1. The leftmost column consists of the slice images in the axial view of the CT scan.

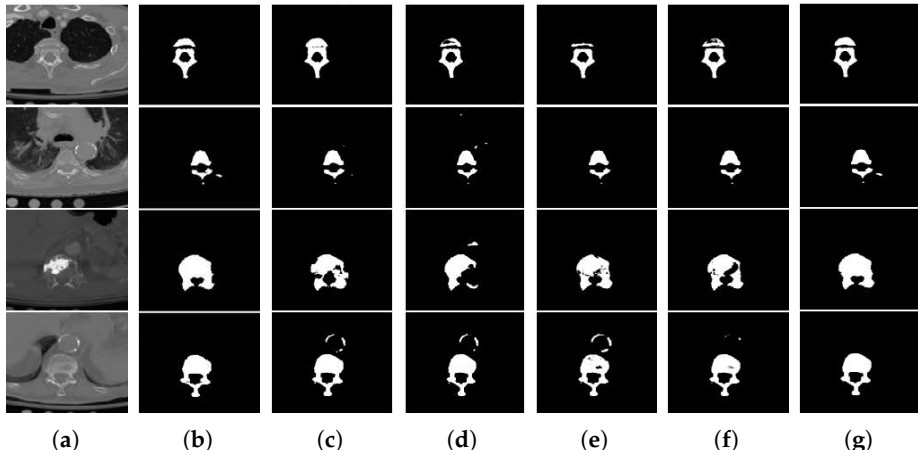

|  (a) | (b) | (c) | (d) | (e) | (f) | (g) |

**Figure 5.** Visual comparison. (**a**) Original image. (**b**) Ground truth. (**c**) SegNet. (**d**) AU-Net. (**e**) U-Net. (**f**) U-Net++. (**g**) Verte-Box.

As shown in the first row of Figure 5, the segmentation results of *SegNet*, *U-Net*, *AU-Net*, and *U-Net++* are under-segmentation in the vertebrae body compared with the ground truth. The second row shows that none of these methods can segment the right transverse process, indicating that these methods are less capable of segmenting smaller objects, while our method can be well aware of and segment these small objects. In the third row, all methods are focused on the lesions area (brighter area) and ignore the real vertebrae, which results in under-segmentation. Only our model has successfully segmented the whole vertebrae. The fourth row is illustrated to show the strong ability of our model in suppressing the background interference (the bright circle above the vertebrae body). Although the performance of *U-Net++* is better than these methods, it is not as good as ours. Our method can exclude the background interference and focus on the real object to be segmented. In general, the above segmentation results show that our model performs best in all cases and shows good segmentation stability.

The quantitative comparison of different methods can be found on Table 1, where all values are mean and standard deviation, and the best performances are bolded. Figure 6 includes the box plots with statistical annotations. Table 2 gives the *p* value of each metric that was calculated using different methods and our model. To vividly illustrate the training process of our model, Figure 7 gives the training and validation *DSC* curves.

**Table 1.** Quantitative comparison of different methods on the test set.

| Method | DSC | IOU | 95% HD |
|--------|-----|-----|--------|
| SegNet | $0.8977 \pm 0.0066$ | $0.8312 \pm 0.0087$ | $12.1561 \pm 1.4937$ |
| AU-Net | $0.8981 \pm 0.0062$ | $0.8377 \pm 0.0092$ | $12.4024 \pm 2.9603$ |
| U-Net | $0.9069 \pm 0.0029$ | $0.8478 \pm 0.0036$ | $10.7243 \pm 0.2378$ |
| UNet++ | $0.9085 \pm 0.0038$ | $0.8511 \pm 0.0042$ | $9.7663 \pm 0.3086$ |
| Ours | $\mathbf{0.9218 \pm 0.0045}$ | $\mathbf{0.8729 \pm 0.0058}$ | $\mathbf{7.7107 \pm 0.5958}$ |

The best performances are bolded. Our model achieved the best results in every metric.

According to the quantitative results shown in Table 1, our model has an improvement of 1.5% in terms of *DSC* and 2.5% in *IOU* over *U-Net*, which means that our model can segment vertebrae accurately from the background. From the point of view of the 95% Hausdorff distance, it also improves the boundary contour effectively, because the smaller the 95% HD value, the more consistent the boundary between the predicted result and the label.

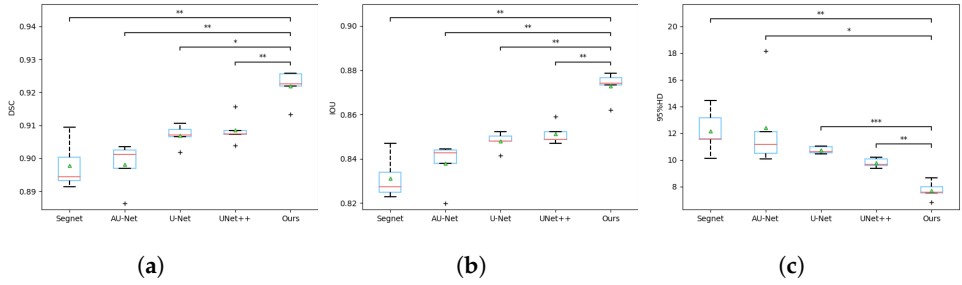

| (a) | (b) | (c) |

**Figure 6.** The box plots of metrics between Verte-Box and other algorithms. (**a**) DSC coefficient comparison; (**b**) IOU coefficient comparison; (**c**) 95% HD coefficient comparison. The central red lines indicate median values, green triangles the average values, boxes the interquartile range, whiskers the smallest and largest values, and data points (+) outliers. * indicates a significant difference between the corresponding experiments, with ** $p \leq 0.01$, and * $p \leq 0.05$ (paired *t*-test).

**Table 2.** *p* value of different methods.

| Method | SegNet-Ours | AU-Net-Ours | U-Net-Ours | UNet++-Ours |
|--------|-------------|-------------|------------|-------------|
| DSC | 0.0065 | 0.0023 | 0.0117 | 0.0089 |
| IOU | 0.0021 | 0.0024 | 0.0039 | 0.0022 |
| 95% HD | 0.0081 | 0.0437 | 0.0002 | 0.0022 |

As shown in the *DSC* box plot of Figure 6a, the experimental results of *SegNet* and *AU-net* are scattered, indicating that their segmentation stability is poor. *U-Net* and *U-Net++* are at a similar level in general. Although the distribution of the results is compact, there are many outliers. Our method has achieved the highest mean value and a suitable data distribution. The *IOU* box plot of Figure 6b is similar to the *DSC* box plot. On the 95% Hausdorff distance (95% HD) box plot of Figure 6c, the mean value of our method is significantly lower than that of the above methods, and the data distribution is more compact.

It can be clearly seen from Table 2 that there is a significant difference between our Verte-Box and other compared methods. For example, U-Net achieves an average *DSC* of $0.9069 \pm 0.0029$, whereas our model achieves $0.9218 \pm 0.0045$ ($p = 0.0117$). For *IOU*, U-Net obtains $0.8478 \pm 0.0036$, and our model obtains $0.8729 \pm 0.0058$ ($p = 0.0039$). For 95%

HD, U-Net obtains 10.7243 ± 0.2378, and Verte-Box obtains 7.7107 ± 0.5958 (*p* = 0.0002). These data show that our Verte-Box is statistically significantly better than other deep learning models.

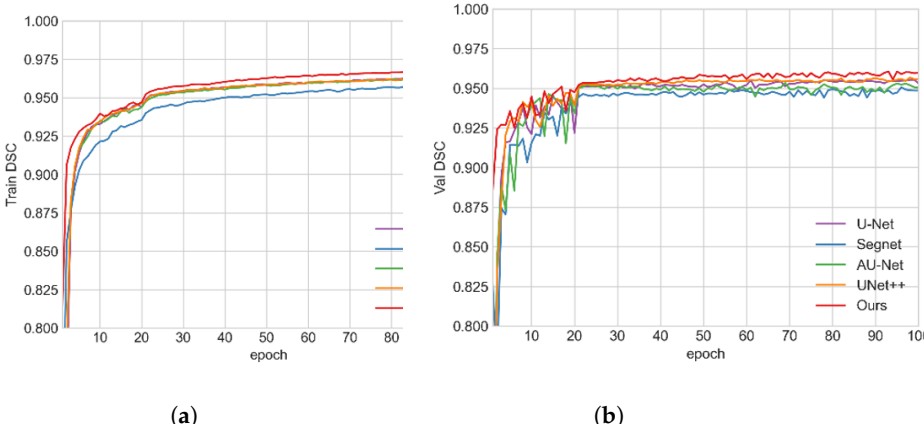

|  |  |
|:---:|:---:|
| (**a**) | (**b**) |

**Figure 7.** Training and validation *DSC* curves of Verte-Box and other algorithms. (**a**) Training curve; (**b**) validation curve.

As shown in Figure 7, in the training process, our *DSC* value is higher than the other algorithms. The training results are in good agreement with the test results. In the validation process, the fluctuation of our model is lowest, proving that our model has a better segmentation stability.

### 3.4. Ablation Study

In order to illustrate the role of the attention mechanism and multi-scale convolution, we have done ablation experiments. Figure 8 gives the segmentation results visualization of different modules. The baseline is *U-Net*. Table 3 shows the quantitative results of the ablation study on the test set; all values are mean and standard deviation. Figure 9 includes the box plots with statistical annotations. Table 4 gives the *p* value of each metric that was calculated using different components and the overall model. *AM* denotes the attention mechanism, and *MSC* denotes multi-scale convolution. *DS* denotes deep supervision.

As shown in columns (d) and (e) of Figure 8, both *AM* and *MSC* improved segmentation results compared with the baseline method. Although the segmentation results of the attention mechanism have small over-segmentation or under-segmentation in the first and fourth row, the presence of multi-scale convolution can eliminate this negative effect, so the overall model performs the best. The second row shows that the attention mechanism can effectively suppress the background interference. The third row demonstrates that multi-scale convolution performs well on small objects. The quantitative results of the ablation study can be found below.

**Table 3.** Quantitative segmentation results of ablation study.

| AM | MSC | DS | DSC | IOU | 95%HD |
|:---|:---|:---|:---:|:---:|:---:|
|  |  |  | 0.9069 ± 0.0029 | 0.8478 ± 0.0036 | 10.7243 ± 0.2378 |
| ✓ |  |  | 0.9107 ± 0.0025 | 0.8566 ± 0.0037 | 9.1597 ± 0.7115 |
|  | ✓ |  | 0.9137 ± 0.0065 | 0.8572 ± 0.0075 | 10.2084 ± 1.4339 |
| ✓ | ✓ |  | 0.9177 ± 0.0039 | 0.8628 ± 0.0037 | 8.7667 ± 0.7739 |
| ✓ | ✓ | ✓ | **0.9218 ± 0.0045** | **0.8729 ± 0.0058** | **7.7107 ± 0.5958** |

The best performances are bolded.

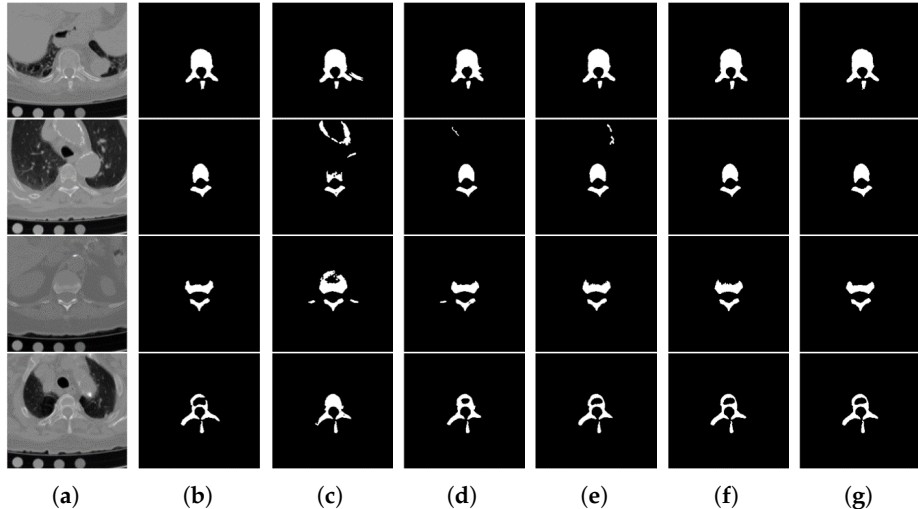

**Figure 8.** Examples from the ablation study. (**a**) Original image. (**b**) Ground truth. (**c**) Baseline. (**d**) AM only. (**e**) MSC only. (**f**) AM+MSC. (**g**) Overall model (AM + MSC + DS).

As shown in the second row of Table 3, a lower 95% HD value demonstrates that the attention mechanism can effectively keep the boundary contour, while the over-segmentation or under-segmentation phenomenon limits its Dice score. The third row shows that the multi-scale convolution plays an important role in improving the accuracy, but a weak anti-interference ability produces a large deviation on the boundary contour of some test images. In the fourth row, we incorporate both into one model, so their strengths become well-balanced, and the overall model performs best. The fifth row shows that deep supervision can further strengthen the segmentation accuracy of the model.

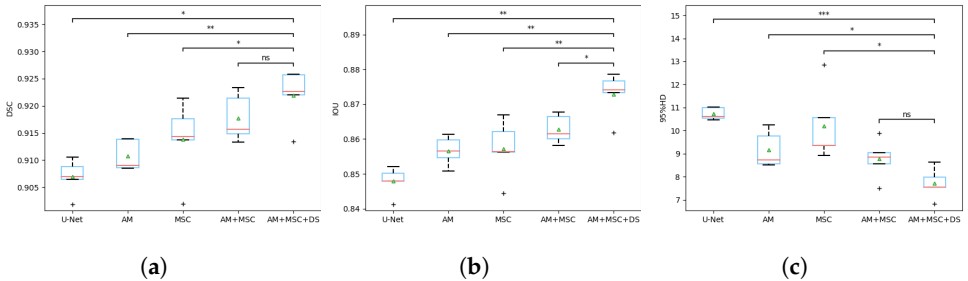

**Figure 9.** The box plots of metrics of ablation study. (**a**) DSC coefficient comparison; (**b**) IOU coefficient comparison; (**c**) 95% HD coefficient comparison. * indicates a significant difference between the corresponding experiments, with *** $p \leq 0.001$, ** $p \leq 0.01$, * $p \leq 0.05$, and ns $p \leq 1$ (paired *t*-test).

**Table 4.** *p* value of different components.

| Method | Baseline | AM | MSC | AM + MSC |
|---|---|---|---|---|
| DSC | 0.0117 | 0.0053 | 0.0137 | 0.1937 |
| IOU | 0.0039 | 0.0078 | 0.003 | 0.0396 |
| 95% HD | 0.0002 | 0.0463 | 0.0294 | 0.0888 |

As shown in the Figure 9a *DSC* box plot, compared with U-Net, both attention mechanism (AM) and multi-scale convolution (MSC) can greatly improve the segmentation result, but the effect of multi-scale convolution is better, indicating its good ability in improving the segmentation accuracy.The combination of the above components achieves better performance, while at the same time, it also leads to a scattered data distribution.

After adding deep supervision (DS), this problem has been significantly alleviated, and the mean value is still improved. From the Figure 9b *IOU* box plot, the effects of the attention mechanism and multi-scale convolution are almost equal, and the improvement after the combination of both is still limited. However, under the role of deep supervision, their advantages have been brought into full play, with a higher mean and closer data distribution. In the Figure 9c 95% Hausdorff distance (95% HD) box plot, the effect of multi-scale convolution is not as good as the attention mechanism, which plays an important role in keeping the boundary contour. The combination of both lays a good foundation for the overall model, whose data are more stable, and the mean value is the highest.

In Table 4, the AM+MSC component achieves an average *DSC* of 0.9177 ± 0.0039, which is worse than that of the overall model (0.9218 ± 0.0045). No significant difference is found ($p = 0.1937$). For *IOU*, AM+MSC obtain 0.8628 ± 0.0037, and the overall model obtains 0.8729 ± 0.0058, with p = 0.0396. For the 95% HD, AM + MSC obtains 8.7667 ± 0.7739, and the overall model obtains 7.7107 ± 0.5958 with no significant difference observed ($p = 0.0888$). In fact, AM+MSC is only one deep supervision away from the overall model, so they have very high similarity.In general, the overall model can generate better segmentation results when compared with the single component.

## 4. Discussion

In this paper, we proposed a novel convolutional neural network to segment vertebrae from computed tomography images. By comparing the performance of the five methods regarding segmentation results using image assessment metrics, it was demonstrated that our model generated superior segmentation predictions compared to the other four methods, with the highest segmentation accuracy and stable segmentation ability.

It is worth noting that the neural network works like a 'black box': all data in this paper are statistically obtained several times. From the box plot of Figure 6, it can be seen clearly that the superiority of our model does not happen occasionally or accidentally. The ablation study demonstrates that the multi-scale convolution and attention mechanism have made a great contribution to the overall model. Multi-scale convolution is responsible for improving the segmentation accuracy, while the attention mechanism plays an important role in keeping the boundary contour. Under deep supervision, their advantages have been brought into full play. Similarly, their role can be verified from the box plot of Figure 9.

The main purpose of this paper is to present a deep-learning-based model, so we only compare it with four related networks. Unlike classification tasks, the annotation for vertebrae segmentation is so difficult and professionally demanding that open access and high-quality segmentation datasets are rare. To our knowledge, CSI 2014 was annotated by doctors and covers the entire thoracic and lumbar spine. Thus, we chose it as the experimental dataset to verify our model. Compared with participating algorithms, our method generated predicted results through only one neural network, so the efficiency is significantly higher. Table 5 gives the comparison between our model and participating algorithms regarding the processing time per case.

**Table 5.** Comparison with participating algorithms.

| Method | Segmentation Strategy | Runtime |
|---|---|---|
| Method 1 [21] | Machine Learning (multi-atlas) | 12 min per case (GPU) |
| Method 2 [22] | Traditional (mean shape model) | 45 min per case (GPU) |
| Method 3 [23] | Traditional (mean shape model) | 10 min per case (GPU) |
| Method 4 [24] | Deep Learning (CNN) | 10 min per case (GPU) |
| Verte-Box | Deep Learning (CNN) | 6 min per case (GPU) |

From the view of expandability, our model seems more similar to a segmentation framework. Obviously, it can be easily transformed into a 3D segmentation network.

Furthermore, both multi-scale convolution and attention mechanisms are expandable. For example, self-attention [25,26] realized a more accurate similarity measurement between two feature representations by dot product calculation. Some studies [27,28] have achieved fairly good results based on this. Although the computational complexity is very high, it is still worth exploring in future work. For multi-scale convolution [29,30], the key point is the parameter. *ASPP* [31] and *PPM* [32], respectively, used atrous convolution and pyramid pooling to capture the local and global information in the image. Thus, multi-scale feature extraction with fewer parameters can also be applied to our model.

### 5. Conclusions

Computed tomography is recognized as a gold standard technique to evaluate spinal disease. Automatic vertebrae segmentation from CT images is critical for spinal disease diagnosis and treatment. In this paper, we presented a novel convolutional neural network *Verte-Box* for the vertebrae segmentation task. Multi-scale convolution was used to extract complex shape features of vertebrae. The interference information contained in the image was suppressed by the attention mechanism that also performs feature representation enhancement at the same time. By experimentation analysis, our model was proven to be better and more stable on segmentation accuracy and stability. Moreover, the method in this paper is quite succinct and more efficient compared with other vertebrae segmentation algorithms. Applying this model to other medical image segmentation tasks and other medical imaging modalities is the next research direction.

**Author Contributions:** Conceptualization, C.L. and B.L.; methodology, C.L., B.L. and S.W.; software, C.L.; validation, C.L., B.L. and G.L.; resources, C.L.; data curation, G.L.; writing—original draft preparation, C.L., B.L. and S.W.; writing—review and editing, C.L., B.L. and S.W.; funding acquisition, B.L. All authors have read and agreed to the published version of the manuscript.

**Funding:** This research was funded by the National Natural Foundation of China under Grant No. 61172167, the Science Fund Project of Heilongjiang Province (LH2020F035), the Youth Science Foundation of Heilongjiang Province (QC2017076), and the Scientific Research Project of Talent Plan of Harbin University of Science and Technology (LGYC2018JC013).

**Institutional Review Board Statement:** Not applicable.

**Informed Consent Statement:** Not applicable.

**Data Availability Statement:** Data are contained within the article.

**Acknowledgments:** The authors would like to thank the instructor Li for her hard work and equipment support, and the laboratory students for their suggestions and comments.

**Conflicts of Interest:** The authors declare no conflict of interest.

### Abbreviations

The following abbreviations are used in this manuscript:

| | |
|---|---|
| DSC | Dice similarity coefficient |
| IOU | Intersection over union |
| 95%HD | 95% Hausdorff 156 distance |
| AM | Attention mechanism |
| MSC | Multi-scale convolution |
| DS | Deep supervision |

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
