# Peer review of "Verte-Box: A Novel Convolutional Neural Network for Fully Automatic Segmentation of Vertebrae in CT Image"

_tomography, doi:10.3390/tomography8010005_

Round 1

Reviewer 1 Report

Thank you for submitting your original paper “CPU-box: A Novel Convolutional Neural Network for Fully Automatic Segmentation of Vertebrae in CT Image”. There is some comments and questions. In this time, these are not all. You describe as “In practice, vertebrae segmentation from volumetric CT image suffers from the following challenges:”, and then you arranged explanation as list. For an example, L1 and L2 is hard to understand because of appearing as words suddenly. Overall, it is lack for detail explanation. Therefore, you should add explanation. In this section, you introduce related works. However, it is difficult for readers to understand the only words. You should explain using symbols and equations. In this journal, “tomography” is keyword. Actually, in this paper, neural network is often used. You should explain relationship between tomography and your work. 

Author Response

Dear reviewer,

Thank you very much for your constructive and rigorous comments, those comments are very helpful for revising and improving our paper. We have studied the comments carefully and made corrections which we hope meet the approval. The revisions are list as follows:

Point 1:  For an example, L1 and L2 is hard to understand because of appearing as words suddenly. Overall, it is lack for detail explanation. Therefore, you should add explanation.

Response 1: According to your comments, we added detail explanation about the technical term to the latest version of manuscript. L1 denotes the first lumbar vertebrae, L2 denotes the second lumbar vertebrae. Details can be found at line 30.

Point 2: In this section, you introduce related works. However, it is difficult for readers to understand the only words. You should explain using symbols and equations.

Response 2: Yes, we try to explain the related work using more symbols and equations, just as above explanation about the challenges. Unfortunately, those methods are not suitable to explain using this way. We have reduced some unnecessary explanations and compressed the introduction section. Details can be found at line 34-62.

Point 3: In this journal, “tomography” is keyword. Actually, in this paper, neural network is often used. You should explain relationship between tomography and your work. 

Response 3: The relationship between this work and tomography was marked up throughout the full text , i.e. the modality of our research object is computed tomography, we aim to analyze those images using deep-learning method. So we proposed a novel neural network Verte-box in this paper. By experimentation analyzing, it has a better performance. Details can be found at line 24-25,266-267.

Thank you again and wish you a happy life.

Best Regards,
Chuang liu

Reviewer 2 Report

The design of study is mostly sound, and manuscript is generally well-written. However, the reviewer has comments on the manuscript.

In Figure1 and 3, there is a need for unity in punctuation. Please unify whether to put a blank space after the parentheses.

The introduction is too long, and it is necessary to introduce the purpose of the paper in the last paragraph. The discussion is relatively short, and some content will have to go into the discussion.

There is no explanation for each abbreviation in figure 2.

You said you evaluated it in the 2014 challenge, how exactly did you prove your excellence? How well was it numerically superior to the models of other participants? And it's already 2021, and new models and papers may have come out, but there is no review for it. There is little explanation as to why this paper should be published. These should be included in the Discussion.

I don't know if the purpose is to present a model or if the purpose is to show that it is superior. Rather than showing the 2014 model, the significance of this paper should be to describe what kind of results it shows by applying it to recent patient data.

I would encourage the authors to clarify why their findings are important, meaningful.

Author Response

Dear reviewer,

Thank you very much for your professional and rigorous comments, those comments are very helpful for revising and improving our paper. We have studied the comments carefully and made corrections which we hope meet the approval. The revisions are list as follows:

Point 1: In Figure1 and 3, there is a need for unity in punctuation. Please unify whether to put a blank space after the parentheses.

Response 1: According to your comments, we have unified the format of figure in this paper. There is a blank after the parentheses.

Point 2: The introduction is too long, and it is necessary to introduce the purpose of the paper in the last paragraph. The discussion is relatively short, and some content will have to go into the discussion.

Response 2:The length of introduction has compressed, and we introduce the purpose in detail in the latest paragraph. We also re-written the discussion section. Details can be found at the introduction section (line 21-75) and the discussion section(line 232-264).

Point 3: There is no explanation for each abbreviation in figure 2.

Response 3: We added the explanation for each abbreviation in figure 2(line 87).

Point 4: You said you evaluated it in the 2014 challenge, how exactly did you prove your excellence? How well was it numerically superior to the models of other participants? And it's already 2021, and new models and papers may have come out, but there is no review for it. There is little explanation as to why this paper should be published. These should be included in the Discussion.

Response 4: First, our method has higher efficiency in processing one case using less time. Table 3 gives the comparison regarding the process time per case. Details can be found at the third paragraph of discussion section(line 246-255).

Point 5: I don't know if the purpose is to present a model or if the purpose is to show that it is superior. Rather than showing the 2014 model, the significance of this paper should be to describe what kind of results it shows by applying it to recent patient data.

Response 5: What we want to talk about in this paper is deep-learning method, while most participating algorithms are traditional or machine-learning-based. In fact, this paper is not focus on comparing with other models or participating algorithms, but to present a deep-learning model. We just want through a comparison with other literature performing the same task on the same challenge to understand the novelty or of the proposed method. CSI 2014 was annotated by doctors and covers the entire thoracic and lumbar spine. So we choose it as the experiment dataset to verify our model.

Point 6: I would encourage the authors to clarify why their findings are important, meaningful.

Response 6: As you said, it will be more persuasive and meaningful by applying our model to recent patient data. Unfortunately, we do not have the recent vertebrae data. but we are planning to apply this model to thyroid nodules segmentation task.

Thank you again and wish you a happy life.

Best Regards,
Chuang liu

Round 2

Reviewer 2 Report

I appreciate the efforts of the authors and can see many improvements. I hope you do a lot of better research.

It would be better if you cite the following papers.

Kim, J.-H.; Mo, Y.-C.; Choi, S.-M.; Hyun, Y.; Lee, J.W. Detecting Ankle Fractures in Plain Radiographs Using Deep Learning with Accurately Labeled Datasets Aided by Computed Tomography: A Retrospective Observational Study. Appl. Sci. 202111, 8791. https://doi.org/10.3390/app11198791

Author Response

Dear reviewer,

Thank you very much for your careful review and encouragement, you have given us a lot of confidence and let us go further on the road of scientific research. We have studied the comments carefully and made corrections which we hope meet the approval. The revisions are list as follows:

Point 1: It would be better if you cite the following papers.

Kim, J.-H.; Mo, Y.-C.; Choi, S.-M.; Hyun, Y.; Lee, J.W. Detecting Ankle Fractures in Plain Radiographs Using Deep Learning with Accurately Labeled Datasets Aided by Computed Tomography: A Retrospective Observational Study. Appl. Sci. 202111, 8791. https://doi.org/10.3390/app11198791

Response 1: This work is very creative and interesting, which proved the powerful ability of deep learning in AI assisted diagnosis and treatment. This paper also gives us a lot of inspiration, so we have cited it in our paper. You can find it at Line 48 and reference 9.

Thank you again and wish you a happy life.

Best Regards,
Chuang liu
